# Molecular Genotyping by 20K Gene Arrays (Genobait) to Unravel the Genetic Structure and Genetic Diversity of the *Puccinia striiformis* f. sp. *tritici* Population in the Eastern Xizang Autonomous Region

**DOI:** 10.3390/plants14101493

**Published:** 2025-05-16

**Authors:** Mudi Sun, Wenbin Chen, Qianrong Yong, Xinyu Kong, Xue Qiu, Jie Zhao

**Affiliations:** 1State Key Laboratory for Crop Stress Resistance and High-Efficiency Production, College of Plant Protection, Northwest A&F University, Yangling 712100, China; sunmudi@nwafu.edu.cn (M.S.); 2020010558@nwafu.edu.cn (W.C.); yongqr@nwafu.edu.cn (Q.Y.); 2310219267@nwafu.edu.cn (X.K.); 2College of Plant Protection, Northwest A&F University, Yangling 712100, China; 2023010410@nwafu.edu.cn

**Keywords:** wheat stripe rust, genetic diversity, genotyping, 20K gene arrays (Genobait), migration pattern

## Abstract

Wheat stripe rust, caused by *Puccinia striiformis* f. sp. *tritici* (*Pst*), poses a significant threat to wheat production in China. Previous epidemic studies have demonstrated the potential of high genetic diversity in the southwest regions of China. Among this epidemic region, the eastern Xizang (Tibet) region holds particular significance, as both wheat and barley crops are susceptible to *Pst*. However, limited information exists regarding the level of population genetic diversity, reproduction model, and migration patterns of the rust in eastern Xizang. The present study seeks to address this gap by analyzing 146 *Pst* isolates collected from the Basu, Zuogong, and Mangkang regions, genotyping by the 20K target Gene Array (Genobait). Our results showed relatively low genotypic diversity in the Basu region, while the highest genetic diversity was observed in the Mangkang area. Structural analysis revealed the abundance of admixed groups in Mangkang, which exhibited this population occurred due to sexual recombination between two different ancestor groups. Gene flow was observed between Zuogong and Basu populations, but it almost did not occur between Mangkang and Zuogong/Basu populations. This region is the world’s highest-altitude epidemic area, thus facilitating the evolution of the rust and possessing the potential to transmit newly evolved *Pst* races to lower wheat-growing regions. Implementing disease management strategies in this area is of potential importance to prevent the transmission of *Pst* races to other parts of Xizang, even neighboring regions possibly. This study facilitates our understanding of epidemiological and population genetic knowledge and the evolution of *Pst* in Xizang.

## 1. Introduction

Wheat stripe rust, caused by a Basidiomycota fungus, *Puccinia striiformis* f. sp. *tritici* (*Pst*), is a significantly destructive disease in wheat-growing regions in China and many countries of the world. This disease can lead to substantial yield reductions, with severe cases resulting in complete crop failure, particularly in highly susceptible wheat cultivars [1,2,3,4,5].

China holds significant importance in the context of *Pst* epidemiology. The favorable environmental conditions annually support the survival of this pathogen. Furthermore, the presence of alternative host species, such as barberry, facilitates the potential evolution of more virulent forms [6]. In China, the Southwest epidemic region played a crucial role in the race evolution and migration of newly evolved races to other regions of the country, especially the Yunnan and Guizhou epidemic areas [7]. A recent study by Awais et al. [8] revealed the regional migration of this pathogen from Central Asia to Xinjiang, China. The study also indicated that this region is potentially suitable for foreign invading races to adapt to the local Chinese environmental conditions. This poses a threat not only to Chinese wheat production but also has the potential to affect global food security due to the pathogen’s long dispersal capabilities. It is important to understand *Pst* population genetic structure in different ecological zones of China, such as the Xizang autonomous region (abbrev., Xizang), where pathogen epidemiology is relatively isolated from other epidemic areas.

Among the various epidemiological zones of China, the eastern Xizang region stands out as the world’s highest-altitude epidemic area. Where wheat is an important food crop in addition to barley, it is grown on approximately 28,100 hectares annually (averaged data in 2015–2023, http://zdscxx.moa.gov.cn:8080/nyb/pc/frequency.jsp (accessed on 24 March 2025)). In this region, stripe rust is a frequently occurring disease on wheat, with annually infected areas of 23,300 hectares accounting for 82.9% of the total growing areas, seriously threatening wheat production. Due to the unique topographic feature of high mountains as barriers and highland climate, Xizang, an isolated stripe rust epidemiological region, is distinct from other inland regions in China. To date, although several studies have investigated the genetic diversity and virulence structure of stripe rust populations in Xizang, most of these studies were conducted in Linzhi County, excluding Zhaluo County in Shannan [9,10,11,12], but not in eastern Xizang. Despite the lack of studies associated with the stripe rust population in eastern Xizang, the epidemics and evolution of *P. striiformis* f. sp. *tritici* in this region remain poorly understood. Therefore, the present study was conducted to gain insights into the genetic diversity and migration patterns of *P. striiformis* f. sp. *tritici* populations in eastern Xizang, with the goal of guiding the management of wheat stripe rust in this region.

## 2. Results

### 2.1. Variant Calling and Data Filtration

A total of 146 samples from the Xizang region of China were sequenced using the 20K gene array GenoBait. The raw data were subsequently filtered to remove homozygous genomic sites and missing data. After filtering, a total of 2060 single nucleotide polymorphisms (SNPs) were obtained for each sample, which were subsequently utilized for genotyping analysis (Table 1). The percentage of available data was 98.4%, with that of missing data being 1.6%. Gametes obtained from filtered data reached 98.4% of the total, and there were only missing gametes at 1.6%. All data of statistic types showed a high sequencing quality, which qualified for data analysis.

### 2.2. Genetic Diversity of Pst in the Eastern Xizang Epidemic Regions

Wheat stripe rust surveillance was conducted in three distinct regions of Xizang, China: Mangkang, Zuogong, and Basu. A high level of gene (*He*) and genotypic diversity was observed in the overall eastern Xizang population (*He* = 0.323, Lambda, *λ* = 0.99; Table 2). Within different regions of eastern Xizang, the lowest *F_hom_* value was observed in the Mangkang region, suggesting the potential for high genetic heterozygosity (Figure 1a). There was no significant difference between expected and observed heterozygosity (Figure 1b), suggesting a sexually recombinant population in this region. We compared the result of gene diversity among sub-regions and found that the Zuogong population exhibited a relatively low level of gene diversity (0.308). While the Mangkang population showed a high gene (0.326) and genotypic diversity (*λ* = 0.98; Table 2; Figure 1c,d). The lowest genotypic diversity was noted in the population of the Basu region (*λ* = 0.96).

### 2.3. Genetic Structure and Population Subdivision Within the Eastern Xizang

The genetic structure of the *Pst* population in eastern Xizang was analyzed using STRUCTURE software (ver. 2.3.4). The *Pst* samples were clustered using the admixed model in Structure software from K = 2 to K = 10. However, the optimal K value (the number of sub-populations in a population) that resulted in population structure subdivision was K = 4 (Figure 2a). Subsequently, K = 4 and higher values did not yield distinct clusters.

Based on the Structure results, we identified three genetic groups (G1, G2, G3, and Admixed group (AD)). These genetic groups were subsequently spatially mapped in the eastern Xizang region of Basu, Zuogong, and Mangkang (Figure 2b). We observed that the overall Basu population consists of the G1 group. This group was also predominant in the Zuogong region. Additionally, this group was limited and scattered as well in the Mangkang. The G2 group was exclusively found in Mangkang, while the G3 group was shared between Mangkang and Zuogong. Notably, we discovered a substantial presence of the Admixed group (AD) in the Mangkang region. This finding implies that this population originated from sexual reproduction between two distinct ancestral populations. These results were also corroborated by principal component analysis (PCA) analysis (Figure 2c,d).

### 2.4. Genetic Divergence

Genetic divergence within the eastern Xizang population was quantified using the *F_ST_* value (Table 3), STRUCTURE analysis (Figure 2a–d), and phylogenetic analysis (Figure 3). The results revealed that the G1 group was abundant between the Zuogong and Basu regions, indicating a close genetic relationship between the Zuogong and Basu *Pst* populations. Furthermore, the low divergence among these two regions was corroborated by the low *F_ST_* value (0.04). Notably, the highest divergence was detected between the Mangkang and Basu (*F_ST_* = 0.151; Table 2), which was also confirmed by the STRUCTURE analysis, suggesting no same genetic group cloned among these two populations.

Unique multi-locus genotypes (MLGs), MLG16, MLG28, MLG35, MLG47, MLG92, and MLG123, were detected in the Mangkang population but not in both the Zuogong and Basu populations. A unique MLG (MLG120) was detected in the Zuogong population but not in the Mangkang and Basu populations. A shared MGL MLG50 was found in Mangkang and Basu populations, and MLG130 was detected in Mangkang and Zuogong populations. The MLG137 was shared in Zuogong and Basu populations (Figure 4).

### 2.5. Migration Pattern

The migration pattern was elucidated through the analysis of genetic group sharing among various regions based on structural analysis, genetic distance (*F_ST_* value), and migration weight using Treemix software ver. 1.13 (Figure 5a–d). We observed that G1 was shared among Zuogong and Basu, indicating a high degree of gene flow between these regions. This was further proofed by the low genetic distance and migration weight values (Figure 5a–d). Comparatively, low gene flow was observed between the Zuogong or Basu population and the Mangkang subpopulations (Mangkang−1, −2, and −3).

## 3. Discussion

Population genetic structure of the eastern Xizang region has not been previously studied, particularly in the areas of Basu, Mangkang, and Zuogong. The present study focuses on conducting wheat rust surveillance in the eastern part of Xizang and revealed high levels of population genetic diversity of *P. striiformis* f. sp. *tritici*. The results facilitate our understanding of the genetic profiles and migration patterns of *P. striiformis* f. sp. *tritici* populations, enabling us to develop effective integrated management strategies for wheat stripe rust control in this region.

Low level of the resistance of wheat varieties to stripe rust and shortage of stripe rust-resistant genes may affect population dynamics of *P. striiformis* f. sp. *tritici*. In Xizang, wheat varieties (lines) and landraces were generally susceptible to stripe rust and carried fewer *Yr* resistance genes. Zhao et al. [13] reported that 72 of 81 wheat landraces grown in Xizang were susceptible to two predominant races, CYR34 and CYR32, except for 3 with seedling stage resistance and 6 with adult plant resistance. Among wheat landraces, three carried *Yr18*, 36 carried *Yr48*, six carried *Yr65*, and two carried *Yr67*. None of the all-wheat landraces carried *Yr5*, *Yr10*, and *Yr24*. Likewise, a previous study by Peng et al. [14] reported that only two stripe rust resistance genes, *Yr15* and *Yr26* (=*Yr24*) out of seven *Yr* genes, including *Yr1*, *Yr5*, *Yr9*, *Yr10*, *Yr15*, *Yr18*, and *Yr26*, were detected in 48 wheat varieties (lines) cultivated in Xizang, with a low frequency of 10.4%. More than half of the tested wheat varieties (lines) were highly susceptible in a two-year field rust test. Additionally, *Yr1*, *Yr3*, *Yr5*, *Yr6*, *Yr8*, *Yr9*, *Yr18*, and *YrA6* are ineffective against local stripe rust populations in Xizang. This situation is endangering the safe production of wheat in Xizang. Additionally, during our field surveys in recent years, we observed that stripe rust infection on wheat cultivars was quite common in Xizang. Thus, the high level of susceptibility of wheat varieties (lines) and landraces in Xizang could be an important factor influencing the component of the Xizang *P. striiformis* f. sp. *tritici* population, and the low level of *Yr* gene richness also may accelerate potentially directional selection of the stripe rust population in Xizang. However, simultaneously, severe stripe rust infection on wheat cultivars grown currently in this region alerts local wheat growers to change wheat cultivars carrying different effective *Yr* genes to the local *Pst* population from those being used at present. On the other hand, due to the low level of wheat resistance to stripe rust locally, it is most important to be aware of introducing stripe rust-resistant wheat cultivars to eastern Xizang and accelerating wheat breeding with more effective *Yr* genes to improve the resistance of local wheat cultivars. These measures can facilitate local wheat growers’ decrease of wheat yield resulting from stripe rust.

Although the level of genetic diversity of *P. striiformis* f. sp. *tritici* populations in Xizang has been assessed previously by different molecular markers, it was inconsistent. Wang et al. [15] reported that low genetic diversity was found in the *P. striiformis* f. sp. *tritici* population in Linzhi. On the other hand, high genotypic diversity of *P. striiformis* f. sp. *tritici* populations was found in Xizang [10,11]. Additionally, it has been demonstrated that genetic variation in relation to genetic divergence took place within the population [15]. Most recently, Du et al. [12] reported that the stripe rust populations in Linzhi exhibited high genetic diversity. Possibly, sexual recombination could contribute to the high level of population genetic diversity of *P. striiformis* f. sp. *tritici* in Xizang. In this study, we observed the high level of genetic diversity of *Pst* in eastern Xizang, especially in Mangkang. We also found the dominant admixed population, which could be established through sexual reproduction between two distant population groups. Although direct evidence on this postulation is still lacking in the present work, based on our field surveys in recent years and this field investigation, we observed that different *Berberis* species are widely distributed in Xizang and usually grow in a patch in some regions. Moreover, six *Berberis* species that distribute in Xizang have been reported to serve as alternate hosts of *P. striiformis* f. sp. *tritici* [16]. Among these *Berberis* species, *B. polyantha* is widely distributed in Xizang and was also observed in some regions of eastern Xizang. More importantly, the sexual cycle of the rust pathogen can occur in this region in spring and autumn [17,18]. Additionally, we observed massive telia covering almost the whole infected leaves of wheat plants in Mangkang and its surrounding regions (Figure 6). This hint of the strong ability of sexual reproduction of the stripe rust pathogen in eastern Xizang is due to high telial reproduction associated with sexual reproduction [19]. The presence of many barberry (*Berberis* spp.) bushes and sexual reproduction of the stripe rust could be responsible for high genetic diversity in Xizang [16,17,18]. Even though considerations for further investigating and demonstrating the occurrence of the sexual cycle of this fungus in eastern Xizang are needed, spraying fungicide on barberry plants around wheat fields at the early pycnial stage (first appearance of light yellow lesions on young leaves of barberry shoots) should be taken into account to reduce the emergence of new races and extend the use of wheat cultivars.

Xizang, located in the Qinghai-Xizang Plateau with an average altitude of >4000 m, is surrounded by high mountains that are geographic barriers that block gene exchange of *P. striiformis* f. sp. *tritici* populations between Xizang and other inland regions. A few studies showed that in the Xizang population structure of *P. striiformis* f. sp. *tritici* was relatively stable, and some races prior to CYR28, such as races CYR10, CYR21 to CYR27, and CYR28, usually can be detected in Linzhi currently, but not in other inland regions of China for decades [9,11,20]. Simultaneously, predominant races that appeared in inland regions have been detected in inland regions for many years but have not been identified in Xizang [9,11,20]. Even though gene flow can occur in different populations within this region [9,15], sometimes there was a greatly significant difference [11], or it occurred extremely limited [11,21]. Therefore, high mountains are likely an important factor responsible for limiting exchange of the rust in Xizang [22]. Spore migration could occur within some regions of Xizang. In the present study, weak gene flow between the Basu and Mangkang regions was observed. This geographic feature affects the rapid evolution of *P. striiformis* f. sp. *tritici* in Xizang, later than inland regions.

Previous studies demonstrated that *P. striiformis* f. sp. *tritici* populations had complicated genetic composition and had unique genetic lineage in Linzhi, Xizang [9,10,11]. In this region, even if the isolates were collected in a sampling site, they were remarkably diverged genetically and belonged to different genetic groups. Moreover, there was a difference in population genetics or population structure in isolates collected in different years in the same region or in those in different regions in the same year [9]. Early studies showed that Xizang is an isolated stripe rust epidemiological region [23,24]. Molecular marker-based analyses demonstrated that the stripe rust population in Xizang had almost complete genetic divergence distinguished from other inland regions, with unique genetic lineage [7,10]. Most recently, a study based on genome level reported the existence of spore migration from Xizang to neighboring provinces, including Qinghai and Sichuan [25]. Even so, however, the role of inoculum from Xizang to both regions has still been absent, postulated to potentially function as a “continent-island” model that needs to be further investigated. Perhaps such spore migration was an occasional incident in an uncertain year in this region. Even if spore migration occurred between Xizang and Qinghai, between Xizang and Sichuan, or among the three regions, a multi-year population genetic study of the stripe rust in these regions and/or neighboring provinces is necessary to confirm spore spread and gain insights into the evolution of the pathogen in Xizang. This study can also facilitate the identification of inoculum sources from Xizang to neighboring provinces in wheat stripe rust.

## 4. Methods

### 4.1. Fungal and Plant Materials

Totally, 146 leaf samples with stripe rust symptoms collected from wheat (cv. unknown) plants at 7 sampling sites in 3 counties, including Basu (26 samples), Mangkang (85 samples), and Zuogong (35 samples) in eastern Xizang in September 2024 (Table 4), were used in this study. Wheat cultivar (cv.) Mingxian 169, highly susceptible to all known Chinese races of *P. striiformis* f. sp. *tritici*, was used to recover isolate and increase spores. Seven-ten seeds of wheat cv. Mingxian 169 were grown in a potting mixture (Inner Mongolian Mengfei Biotech Co., Ltd., Inner Mongolia, China) in a pot and cultivated in a greenhouse with a dual photoperiod system of 16 h light/8 h darkness at 20–25 °C until ten-day-old seedlings were ready for use.

### 4.2. Pure Isolate

A pure isolate of a stripe rust leaf sample was established by picking an individual uredium to inoculate a dewaxed leaf of ten-day-old seedlings of wheat cv. Mingxian 169. After spraying with an atomizer, inoculated wheat plants were covered using a transparent plastic column rolled by a polystyrene film (Deli Group Co., Ltd., Ningbo, China) to avoid cross-contamination and then moved into a dew chamber (I-36 D, Percival Scientific, Inc., Perry, IA, USA) for 24 h at 10 °C in the dark. After incubation, the plants were transferred to a rust-free growth chamber in the greenhouse with temperature and photoperiod conditions of 16 h light/8 h dark at 16 °C/13 °C. As the appearance of infected flecks, a leaf was kept and others were excised. The plants were kept in the growth chamber until sporulation. Fresh urediospores were collected and stored after drying. When a leaf was fully infected, the leaf was cut off, put into a paper bag, and then maintained in a desiccator after complete drying at room temperature until use.

### 4.3. DNA Extraction and Genotyping

A dried leaf segment (~5 cm long) with full−infection by a single−uredium was put into a 2-mL centrifuge tube with three steel beads (4 mm in diameter) and made into a fine powder at a frequency of 45 Hz per minute for 2 min using a tissue lyser (Tissuelyser-48L, Shanghai Jiangxi Industrial Development Co., Ltd., Shanghai, China). The powder was used for genomic DNA extraction by the CTAB method with modification [26]. The quality and quantity of total DNA were measured using a spectrometer (NanoDrop 2000, ThermoFisher Scientific, Waltham, MA, USA). The DNA was preliminarily handled with a GenoBaits End Repair Kit, added sequencing adaptors, and was conducted for demanding purification through multiple steps. The 20K genotyping-by-target sequencing (GBTS) chip for *P. striiformis* f. sp. *tritici* was used for sequencing overall samples. This chip was developed based on the genome of the isolate Pst134E36_v1_pri by Dr. Qingdong Zeng (Northwest A&F University, Yangling, China) after comparing genomes of worldwide *P. striiformis* f. sp. *tritici* isolates released. Sequencing was performed by Shijiazhuang MolBreeding Biotechnology Co., Ltd. (Shijiazhuang, China).

### 4.4. Variant Calling and Filtration at the Genome Level

Raw sequencing data were used for variant calling and filtering with software including UnifiedGenotyper and VariantFiltration of GATK v3.5-0 [27]. To ensure high confidence single nucleotide polymorphisms (SNPs) of each of the overall DNA samples, VCFtools v0.1.16 [28] was carried out with the command “--keep-filtered PASS” to reject variants designated as “HARD_TO_VALIDATE”, with the description “QD < 2.0 || MQ < 40.0 || FS > 60.0 || SOR > 3.0”. Subsequently, the rest of the variants of each DNA sample were taken into a merged population VCF file by running BCFtools v1.18 [29]. The missing rate for these SNPs in the population VCF file was confirmed by filtering with PLINK v1.9 (https://zzz.bwh.harvard.edu/plink/ (accessed on 10 March 2025)), of which parameters “--mind 0.1” and “--geno 0.1” were executed.

### 4.5. Data Analysis

Population genetic diversity was evaluated by calculating the observed number of alleles (*Na*), effective number of alleles (*Ne*), Shannon’s information index (*I*), expected heterozygosity (*He*), and observed heterozygosity (*Ho*). These relevant indexes were calculated by running the software POPGENE v1.32 [30]. The parameters set of poppr in the R language package include the number of multi-locus genotypes (MLGs); the number of expected MLGs (eMLGs) based on rarefaction; Stoddart and Taylor’s index (*G*), which is used for assessing genotypic richness [31]; evenness (E.5), which can be used for predicting the distribution of genotype abundance [32,33,34]; *Nei’s* gene diversity index (*H*), which is computed based on the proportion of different genes among populations [35]; Simpson’s index (*λ*); and Shannon’s information index, which is used for calculating population richness and uniformity [36,37].

Population genetic structure was determined using a model-based Bayesian method with STRUCTURE V.2.3 software [38]. Genotypic data from sequencing by 20K-GBTS chip were used to calculate each K value (K = 2 to K = 4) through 10 independent runs of 100,000 iterations per run, with a burn-in period of 100,000 iterations. Cluster matching was achieved by CLUMPP v1.1.2 and Distruct v.1.1. The output of CLUMPP was visualized to graph exhibiting population structure [39].

## Figures and Tables

**Figure 1 plants-14-01493-f001:**
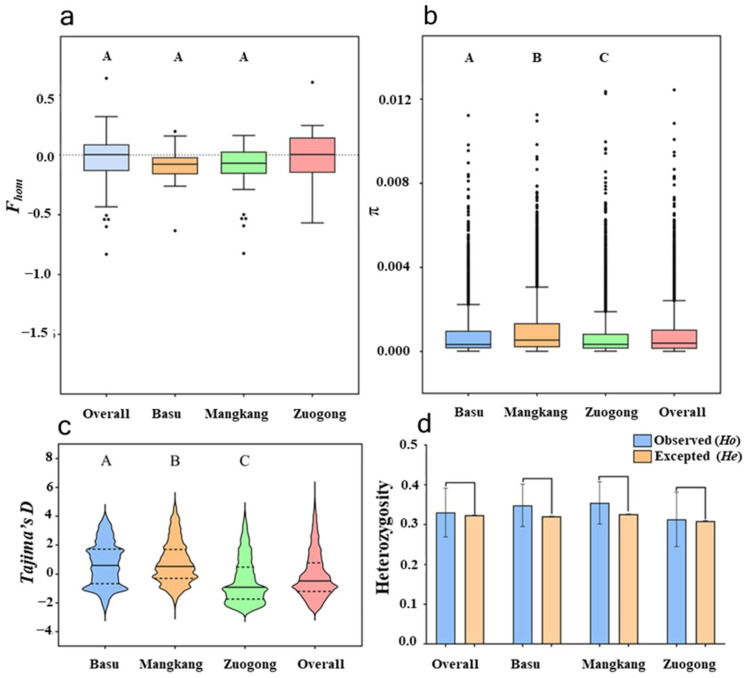
Different diversity parameters of the *Puccinia striiformis* f. sp. *tritici* populations of the eastern Xizang region during 2024 cropping season. (**a**). *F_hom_* is used for evaluating the heterozygosity level. (**b**) *π* of the molecular group (G) significance test was implemented with Kolmogorov–Smirnov. (**c**) *Tajima’s* D, a neutrality test for estimating the heterozygosity level. (**d**) Expected and observed heterozygosity frequency based on a nonparametric Kolmogorov–Smirnov test.

**Figure 2 plants-14-01493-f002:**
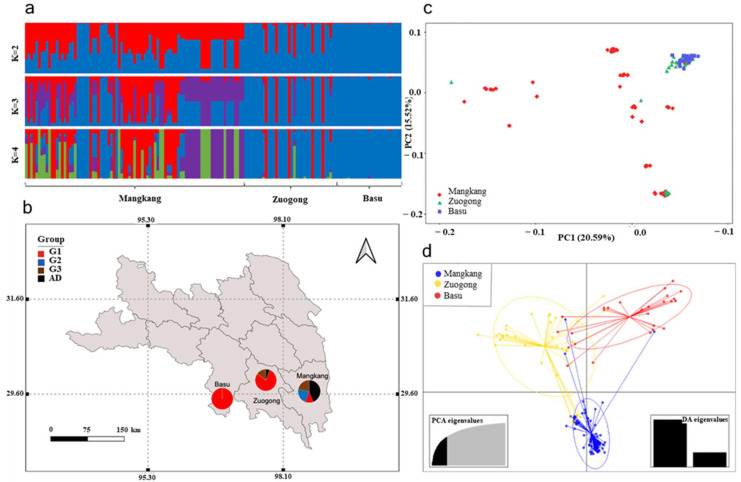
Population genetic structure of *Puccinia striiformis* f. sp. *tritici* isolates collected from different locations in eastern Xizang during the 2024 cropping season. (**a**) Structure analysis results from the number of populations (K = 2 to K = 4). (**b**) Spatial plots illustrating different genetic groups (G1 to G3) and admixture group (AD) of all populations. (**c**) PCA plot showing the distribution of different individual genotypes of populations. (**d**) Principal coordinate analysis (PCA) eigenvalues of different populations.

**Figure 3 plants-14-01493-f003:**
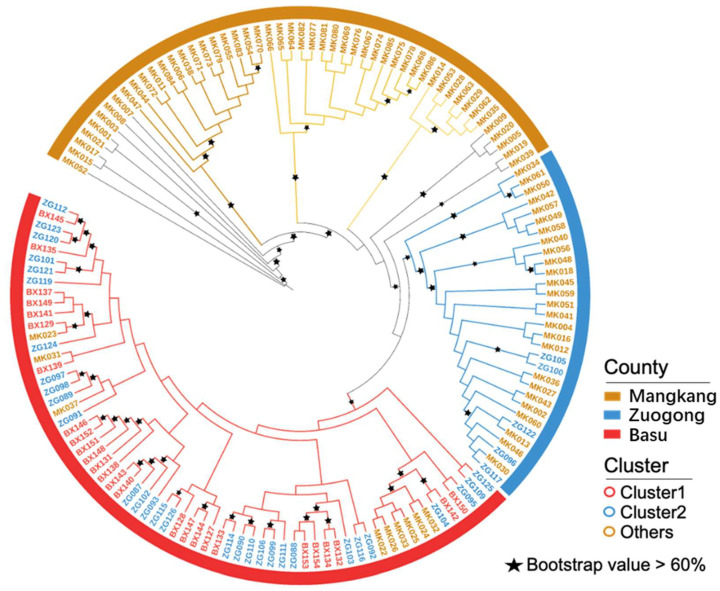
Phylogenic tree showing different individual genotypes of *Puccinia striiformis* f. sp. *tritici* isolates collected from different regions of eastern Xizang during the 2024 cropping season.

**Figure 4 plants-14-01493-f004:**
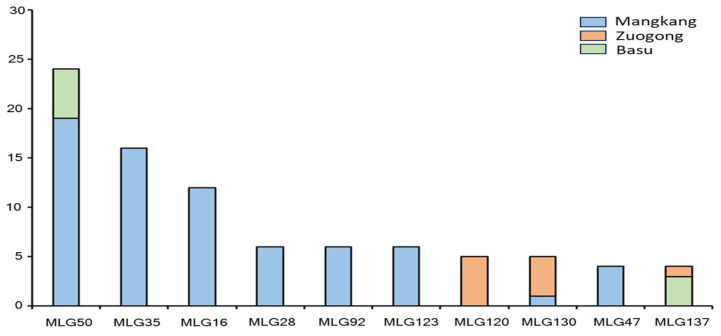
Unique and shared multi-locus genotypes (MLGs) detected in the Mangkang, Zuogong, and Basu populations.

**Figure 5 plants-14-01493-f005:**
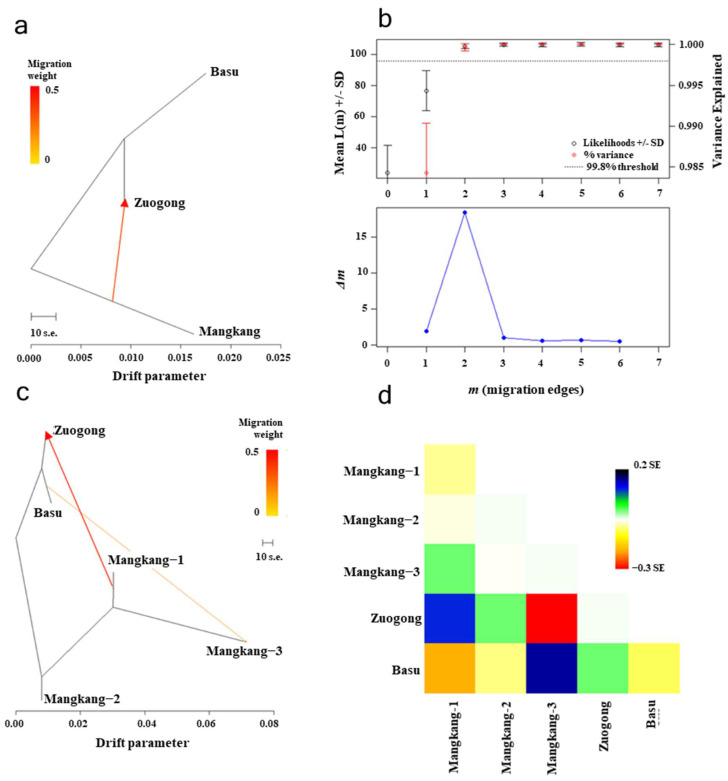
Migration rate of *Puccinia striiformis* f. sp. *tritici* (*Pst*) populations among different locations (Basu, Mangkang, and Zuogong) of eastern Xizang based on detection with TreeMix analysis. (**a**) Migration weight of *Pst* populations of the eastern Xizang region. (**b**) The optimal m by second-order rate of change (Δ*m*) selected from the number of migration events (m). (**c**) The scenario of migration of *Pst* populations within eastern Xizang (Mangkang subpopulations, Mangkang−1, −2, and −3). (**d**) Relatedness of *Pst* populations in the eastern Xizang region through heatmap.

**Figure 6 plants-14-01493-f006:**
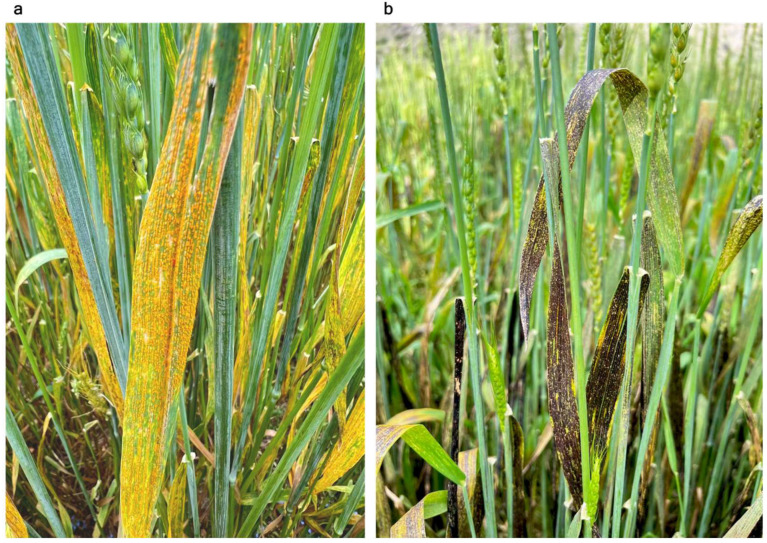
Massive uredia (urediospores) and telia (teliospores) covering almost all parts of infected leaves were observed around Mangkang in eastern Xizang in early September 2024. (**a**) Yellow uredia. (**b**) Black telia.

**Table 1 plants-14-01493-t001:** Overall data summary of *Puccinia striiformis* f. sp. *tritici* isolates used for genotyping.

Stat. Type	Value
Number of taxa	146
Number of sites	2906
Sites × Taxa	424,276
Number not missing	417,591
Proportion not missing	98.4%
Number missing	6685
Proportion missing	1.6%
Number gametes	848,552
Gametes not missing	835,182
Proportion gametes not missing	98.4%
Gametes missing	13,370
Proportion gametes missing	1.6%
Number heterozygous	138,299
Proportion heterozygous	32.6%
Average minor allele frequency	24.1%

**Table 2 plants-14-01493-t002:** Different genetic diversity parameters of *Puccinia striiformis* f. sp. *tritici* populations in eastern Xizang during the 2024 cropping season.

Population	No. of Isolate	Genotypic Richness and Evenness ^a^			Genotypic Diversity ^b^				Index of Association ^c^	
		MLG	E.5	eMLG	*H*	*G*	*lambda*	*He*	*Ia*	*rBarD*
Mangkang	85	85	1	26	4.44	85	0.988	0.326	23.1	0.0144
Zuogong	35	35	1	26	3.56	35	0.971	0.308	31.4	0.0187
Basu	26	26	1	26	3.26	26	0.962	0.32	15.7	0.0126
Total	146	146	1	26	4.98	146	0.993	0.323	23.2	0.0124

^a^ MLG = multi-locus genotype; E.5 = Pielou’s evenness index; eMLG = expected multi-locus genotype. ^b^ *H* = Shannon–Wiener diversity index; *G* = Stoddart and Taylor’s diversity index; *Lambda* = Simpson’s diversity index; *He* = heterozygosity. ^c^ *Ia* = The index of association; *rBarD* = accounts for the number of loci sampled that is less biased. A *rBarD* value is approached to be zero (0) if a population is sexually reproduced.

**Table 3 plants-14-01493-t003:** Pairwise genetic divergence (*F_ST_*) values between *Puccinia striiformis* f. sp. *tritici* populations collected from different regions in eastern Xizang during the 2024 cropping season.

Population	*F_ST_* Value ^a^		
	Zuogong	Basu	Mangkang
Zuogong	0.0000		
Basu	0.0424	0.0000	
Mangkang	0.0701	0.1151	0.0000

^a^ 0 < *F_ST_* < 0.05, low genetic divergence; 0.05 ≤ *F_ST_* < 0.15, moderate genetic divergence; 0.15 ≤ *F_ST_* < 0.25, high genetic divergence; *F_ST_* ≥ 0.25, extremely high genetic divergence.

**Table 4 plants-14-01493-t004:** Collection of *Puccinia striiformis* f. sp. *tritici* isolates from eastern Xizang in the September 2024 cropping season.

Sampling Location		No. of Wheat Fields	No. of Sampling Sites	No. of Isolates
Mangkang	Pula village, Zhatuo town	7	21	20
	Kajun village, Rumei town	5	11	16
	Dangzuo village, Luoni town	9	18	26
	Liemugang village, Cuowa town	7	14	23
Sub-total		28	64	85
Zuogong	Zeba village, Wangda town	5	15	20
	Deliebi village, Tiantuo town	8	13	15
Sub-total		13	28	35
Basu	Zhongba village, Ranwu town	6	12	26
Sub-total		6	12	26

## Data Availability

All data used in this study are available if necessary.

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
