# Peer review of "Molecular Genotyping by 20K Gene Arrays (Genobait) to Unravel the Genetic Structure and Genetic Diversity of the Puccinia striiformis f. sp. tritici Population in the Eastern Xizang Autonomous Region"

_plants, 2025, doi:10.3390/plants14101493_

Round 1
Reviewer 1 Report
Comments and Suggestions for Authors
The manuscript ‘Molecular Genotyping by 20K Gene Arrays (Genobait) to Unravel the Genetic Structure and Genetic Diversity of Puccinia striiformis f. sp. tritici Population in the East Xizang Autonomous Region’ examines how regional differences and conditions influence the population structure of the P. striiformis population. It highlights the variations in factors across different regions, which impact the genetic structure of P. striiformis and the epidemiology of stripe rust in those areas. The manuscript requires major revision.
My comments are as follows.
Abstract:
Line 14: Regions instead of ‘region’
Lines 19-20; Combine two sentences
Lines 22-24: The sentence is not clear
Lines 27-28: Awkward sentence
Introduction:
Line 45: In ‘Pathogen migrating’, ‘migrating’ is redundant.
Line 50: Such as instead of ‘as such’
Lines 51-52: Awkward sentence
Line 57-59: Please revise
Lines 63-64: Please revise
Line 66: Please use ‘autonomous region’ instead of ‘Autonomous Region’ throughout the manuscript.
Results:
Table 1: Please change the formatting of the table. What does value mean?
Line 83: region/regions
Lines 84 and 86: Suggesting instead of ‘suggested’
Line 85: There was nonsignificant observed/There was no significant difference
Line 86: remove ‘the signs of’, please revise the sentence
Lines 87- 88: Please revise
Figure 1: Please define the Y-axis legends in the figure description. For example, what does Tajima’s D mean?
Table 2: In the second column, No. ‘85.’ Should not be bold and remove the period after 85
Line 102: Please revise
Lines 106-108: Explain what the K value is.
Figure 2: Figures and legends are too small. There are two figures c and no figure b.
Table 3: The heading of the first column should be regions not FsT value
Figure 3 and 4: Figures are small. migration event instead of ‘migrate events number’. What is migration edge? What is Δm value?
Discussion:
Line 162: Please provide information about the landraces or wheat lines in the materials and methods. Specify if the lines were spring or winter wheat.
Line 164: susceptible to instead of ‘susceptible for’
Line 166: Remove ‘overall’
Line 167: Please revise
Line 170: Half of the tested instead of ‘A half of tested’
Line 173: Remove ‘in investigated fields’
Lines 178-179 and Lines 182-183: Please revise
Line 181: Genetic richness or diversity?
Lines 182-183: Please revise
Line 184: remove ‘mostly’
Line 184: How close is Linzhi to the study sites in this population?
Line 191: Remove ‘bushes’
Line 199: Remove ‘this’
Line 201: Remove ‘the existence of’
Line 207: barriers instead of ‘barrier’
Line 210-213: Please revise.
Line 216: What is the intensity level?
Lines 219-222: Please revise
Line 222: Remove ‘greatly’
Line 231: Genetic divergence instead of genetic ‘diverge’
Line 232: Remove ‘most recently’ or ‘although’
Lines 234-235: Please revise
Line 241: ‘in the occurrence of’ is redundant
Line 245: Which wheat plants?
Line 252: Greenhouse or growth chamber?
Lines 276-281: Long sentence, please revise
Line 294: Please revise
Author Response
Comment 1:
Abstract:
Line 14: Regions instead of ‘region’
Response: We corrected.
Lines 19-20; Combine two sentences
Response: Agree. We made it.
Lines 22-24: The sentence is not clear
Response: We revised it for making it clear.
Lines 27-28: Awkward sentence
Response: We re-organized this sentence.
Introduction:
Line 45: In ‘Pathogen migrating’, ‘migrating’ is redundant.
Response: Agree. We deleted.
Line 50: Such as instead of ‘as such’
Response: We corrected.
Lines 51-52: Awkward sentence
Response: We rewrote this sentence.
Line 57-59: Please revise
Response: We revised and changed the number in format in line 54.
Lines 63-64: Please revise
Response: We revised.
Line 66: Please use ‘autonomous region’ instead of ‘Autonomous Region’ throughout the manuscript.
Response: Agree. We made it change throughout the manuscript.
Results:
Table 1: Please change the formatting of the table. What does value mean?
Response: We changed, and added the text from the information of the table.
Line 83: region/regions
Response: We corrected.
Lines 84 and 86: Suggesting instead of ‘suggested’
Response: We made it correction.
Line 85: There was nonsignificant observed/There was no significant difference
Response: We changed.
Line 86: remove ‘the signs of’, please revise the sentence
Response: Agree. We deleted it and revised.
Lines 87- 88: Please revise
Response: We revised.
Figure 1: Please define the Y-axis legends in the figure description. For example, what does Tajima’s D mean?
Response: Agree. We revised.
Table 2: In the second column, No. ‘85.’ Should not be bold and remove the period after 85
Response: Agree. We changed.
Line 102: Please revise
Response: We made revision.
Lines 106-108: Explain what the K value is.
Response: We added in the text.
Figure 2: Figures and legends are too small. There are two figures c and no figure b.
Response: We replaced of original Figures in Figure 2.
Table 3: The heading of the first column should be regions not FsT value
Response: We changed.
Figure 3 and 4: Figures are small. migration event instead of ‘migrate events number’. What is migration edge? What is Δm value?
Response: We used a new figure. Disagree change ‘migrate events number’ to be “migrate event”, and changed “migrate” as “migration”. We added figure legends on these two parameters.
Discussion:
Line 162: Please provide information about the landraces or wheat lines in the materials and methods. Specify if the lines were spring or winter wheat.
Response: According to the reference 14, from which information were obtained. We attached the table as a supporting file for reviewing.
Line 164: susceptible to instead of ‘susceptible for’
Response: We changed.
Line 166: Remove ‘overall’
Response: We deleted.
Line 167: Please revise
Response:
Line 170: Half of the tested instead of ‘A half of tested’
Response: We revised.
Line 173: Remove ‘in investigated fields’
Response: We deleted.
Lines 178-179 and Lines 182-183: Please revise
Response: We revised.
Line 181: Genetic richness or diversity?
Response: Genetic diversity. We changed the term in the following sentences.
Lines 182-183: Please revise
Response:
Line 184: remove ‘mostly’
Response: We deleted.
Line 184: How close is Linzhi to the study sites in this population?
Response:
Line 191: Remove ‘bushes’
Response: We deleted.
Line 199: Remove ‘this’
Response: We deleted.
Line 201: Remove ‘the existence of’
Response: We deleted.
Line 207: barriers instead of ‘barrier’
Response: We corrected.
Line 210-213: Please revised and rewrote.
Response: We revised.
Line 216: What is the intensity level?
Response: We deleted ‘in the intensity levlel’
Lines 219-222: Please revise
Response: We rewrote this part.
Line 222: Remove ‘greatly’
Response: We deleted.
Line 231: Genetic divergence instead of genetic ‘diverge’
Response: We corrected.
Line 232: Remove ‘most recently’ or ‘although’
Response: We deleted the word ‘although’.
Lines 234-235: Please revise
Response: We rewrote this sentence.
Line 241: ‘in the occurrence of’ is redundant
Response: We deleted.
Line 245: Which wheat plants?
Response: Sorry for wheat cultivars that are unknown. Possibly, wheat cultivars grown in Xizang are usually bred using a German wheat variety Feimai introduced into Xizang in the 1980s. But Feimai wheat has been susceptible for decades. Herein, we had to note wheat cv. unknown in the text.
Line 252: Greenhouse or growth chamber?
Response: We deleted “in a growth chamber”.
Lines 276-281: Long sentence, please revise
Response: We separated this part into several short sentences.
Line 294: Please revise
Response: We re-organized.

Reviewer 2 Report
Comments and Suggestions for Authors
In this study, authors looked at the genetic makeup of a fungus called Puccinia striiformis f. sp. tritici (Pst). This fungus causes stripe rust, a disease that affects wheat, in the East Xizang (Tibet) area of China. The authorss used a new method, 20K target Gene Arrays (Genobait), to study 146 samples of the fungus collected from three places: Basu, Zuogong, and Mangkang. They found that the fungus was genetically different in each area. Mangkang had the most diverse fungus types, and there was evidence of the fungus reproducing sexually there. They also saw that the fungus was spreading between Zuogong and Basu, but not as much to Mangkang. This study helps us understand how stripe rust spreads and changes in this unique, high-altitude region. The study is good overall, but it needs some revisions before it can be published.
Major comments
Authors talked a lot about the genetic differences they found, but they need to explain how this information can help farmers manage the disease better. They mention disease management in the beginning, but don't give specific advice based on their findings.
Authors need to explain why they chose to study 146 samples of the fungus. They need to show that this number of samples is enough to accurately represent the fungus types in each area.
This study only looks at genetic differences, but it doesn't say anything about how strong or dangerous the different types of fungus are. Knowing this would give a better picture of the stripe rust problem in the region.
Authors think that the diverse fungus types in Mangkang are because the fungus is reproducing sexually. While their data suggests this, they need more proof, like finding parent types of the fungus or studying genes related to reproduction.
Minor comments
Some of the pictures, especially Figure 2b, are hard to understand. Authors need to make them clearer by improving the resolution and adding better labels.
The image quality needs to be high for publication.
Please provide well-described legends for all figures and tables.
There are some small mistakes in the writing, like typos and grammar errors. Authors should proofread carefully.
This study uses different names for the same place ("East Xizang" and "eastern Xizang Autonomous Region"). They should pick one name and use it consistently.
Author Response
Comment 2
Major comments
Authors talked a lot about the genetic differences they found, but they need to explain how this information can help farmers manage the disease better. They mention disease management in the beginning, but don't give specific advice based on their findings.
Response: Thank you for your valuable comments. We added the text about this concern in discussion section.
Authors need to explain why they chose to study 146 samples of the fungus. They need to show that this number of samples is enough to accurately represent the fungus types in each area.
Response: We added Table 4 providing sampling information in details for easy know isolates that were used in this study are of representative.
This study only looks at genetic differences, but it doesn't say anything about how strong or dangerous the different types of fungus are. Knowing this would give a better picture of the stripe rust problem in the region.
Response: Thank you for the comments. I added the text about this concern in discussion section. Also, we provided a picture on stripe rust infection observed in a wheat field in the same time.
Authors think that the diverse fungus types in Mangkang are because the fungus is reproducing sexually. While their data suggests this, they need more proof, like finding parent types of the fungus or studying genes related to reproduction.
Response: Thank you for your critical and valuable comments. In our laboratory, a couple of previous studies by Du et al. [18], and Wang et al. [17] demonstrated the stripe rust fungus can infect alternate (secondary) host barberry to complete life cycle under natural conditions in Tibet. On the other hand, a previous study by Ali et al. [19], entitled ‘ Reduction in the sex ability of worldwide clonal populations of Puccinia striiformis f. sp. tritici’, testified that capacity of telial production is associated with ability of sexual reproduction of the fungus. In China, direct and indirect evidences reported by many studies have demonstrate Chinese Pst populations are sexually reproduced in fields, especially Northwestern and Southwestern regions of the country, including Xizang. In the present study, we did not provide direct evidence about this focus, but we postulated that in the east Xizang, sexual reproduction could occur. We will further investigate to support this postulation in future but not in this study.
Minor comments
Some of the pictures, especially Figure 2b, are hard to understand. Authors need to make them clearer by improving the resolution and adding better labels.
Response: Agree. We revised figure legend in details for easy understanding.
The image quality needs to be high for publication.
Response: Agree. We provided pictures with high resolution and clear text to the editor, and change bad quality pictures in the manuscript.
Please provide well-described legends for all figures and tables.
Response: Agree. We adopted the suggestions and made more information in details for figures and tables.
There are some small mistakes in the writing, like typos and grammar errors. Authors should proofread carefully.
Response: Thank you careful reviewing. We checked and make correction across the manuscript.
This study uses different names for the same place ("East Xizang" and "eastern Xizang Autonomous Region"). They should pick one name and use it consistently.
Response: Agree. We change to a uniform abbreviated name of Xizang autonomous region throughout the manuscript.

Round 2
Reviewer 1 Report
Comments and Suggestions for Authors
Dear authors,
I hope this message finds you well. I am writing to confirm that my comments have been successfully addressed.
Best regards,
Reviewer 2 Report
Comments and Suggestions for Authors
Authors have diligently carried out revisions based on the reviewer's comments.
Now the manuscript is suitable for publication as it is.
It is a well prepared manuscript.